# Analysis of the Sagittal Root Position of the Maxillary and Mandibular Anterior Teeth in the Alveolar Bone Using Cone-Beam Computed Tomography

**DOI:** 10.3390/diagnostics14232756

**Published:** 2024-12-06

**Authors:** Rawa Jamal Abdul, Darwn Saeed Abdulateef, Ara Omer Fattah, Ranjdar Mahmood Talabani

**Affiliations:** 1Oral Diagnosis Department, College of Dentistry, University of Sulaimani, Sulaimani 46001, Iraq; rawa.abdul@univsul.edu.iq; 2Conservative Department, College of Dentistry, University of Sulaimani, Sulaimani 46001, Iraq; darwn.abdulateef@univsul.edu.iq; 3Paedodontic and Community Oral Health Department, College of Dentistry, University of Sulaimani, Sulaimani 46001, Iraq; ara.fatah@univsul.edu.iq

**Keywords:** sagittal root position, immediate implant placement, CBCT, alveolar bone thickness

## Abstract

**Background/Objectives:** The purpose of this study was to measure the bone thickness and angulation of the maxillary and mandibular anterior teeth on the buccal and palatal/lingual sides and also to analyze the sagittal root position (SRP) in the alveolar bone in relation to age and gender using cone-beam computed tomography (CBCT) in an Iraqi subpopulation. **Methods:** CBCT images of 1200 maxillary and mandibular central and lateral incisors and canines from 100 patients (48 males and 52 females) were retrospectively analyzed. These patients were categorized by age into group I ≤ 25, group II 26–40, and group III ≥ 41 years old. The SRP in the alveolar bone was classified as class I, class II, class III, and class IV, and the buccal type was further classified into subtypes I, II, and III. In addition, the buccolingual inclination of the tooth and buccal/palatal/lingual bone thickness at the coronal, middle, and apical thirds were evaluated and then compared based on age and gender. The data were analyzed using the Pearson chi-square test. Descriptive statistics, Kruskal–Wallis and Mann–Whitney U test were used to compare the thickness and angulation according to the SRP classes. **Results:** The mean frequency distribution of SRP of maxillary anterior teeth indicated that most of them were located buccally and were classified as (Class I) and subtype (III). Moreover, for mandibular anterior teeth, the majority were classified as (Class IV) and subtype (II). The mean sagittal angulation of maxillary anterior teeth approximately ranged from 5.9 for tooth 12# to 8.2 for teeth 13# and 23#, while for mandibular anterior teeth it ranged from 7.4 for tooth 33# to 10.3 for tooth 41#. The thickness of bone in the apical third of the buccal side of all maxillary and mandibular teeth was significantly related to age (*p* < 0.05). In the middle third, the thickness of bone in the buccal and palatal side of all maxillary anterior teeth and in the apical third of most mandibular teeth in the lingual side was significantly higher in males (*p* < 0.05). **Conclusions:** A majority of the maxillary anterior tooth roots were positioned close to the buccal cortical plate, while most of the mandibular anterior teeth were engaging both the buccal and lingual cortical plates. Males had more alveolar bone thickness for both maxillary and mandibular anterior teeth, and only the apical portion significantly changed with age. CBCT of the buccal and palatal/lingual bone and SRP is recommended for the selection of the appropriate treatment approach and implant placement.

## 1. Introduction

The placement of the implant plays a crucial role in ensuring both the esthetic appearance and functional longevity of the implant [1]. If the implant is inserted into the extraction socket at the same angle as the root within the jawbone, the prosthetic crown would achieve an optimal position [2,3].

Evaluating the sagittal root position (SRP) of the alveolar bone aids in effective treatment planning, which in turn promotes success and prevents complications. The size, position, and angle of the root typically determine the dimensions of the buccal side of the alveolar bone [4].

Several concerns are related to implant placement in the maxillary anterior area [5,6,7]. Patients demand quick treatment and excellent esthetic results. However, clinicians face challenges in treating this area due to: (i) the labial bone’s tendency to remodel extensively; (ii) the narrow labio-oral dimension of the ridge; and (iii) facial concavity. After tooth removal, the tooth-supporting bone begins to resorb, potentially resulting in a significant bone loss. In cases where the ridge is narrow, this resorption may require additional bone augmentation. Furthermore, managing facial concavity is crucial, especially for implant placement guided by esthetic considerations [8,9,10].

In addition, mandibular anterior alveolar bone presents significant challenges for diagnosis and treatment due to its anatomical features and relatively small dimensions. Furthermore, its outer layer (vestibular cortex) is prone to resorption, which raises the risk of perforation during immediate implant placement [4].

Therefore, preoperative assessment of SRP is essential for ensuring the long-term success of implants in the area of maxillary and mandibular anterior.

To ensure the accurate assessment of SRP through precise radiographic analysis, the American Academy of Oral and Maxillofacial Radiology advocates for CBCT. CBCT is valuable for assessing patient anatomy and determining the feasibility of implant placement by evaluating the adequacy of alveolar bone in providing stable support [11,12,13].

Thus, the aim of this study was to assess and compare the SRP and angulation of maxillary and mandibular anterior teeth for implant placement in an Iraqi subpopulation using CBCT. The null hypothesis is that no association will be detected between buccal and palatal/lingual alveolar bone thicknesses in different areas of anterior tooth roots and the angle of roots in alveolar bone in relation to age and gender.

## 2. Materials and Methods

### 2.1. Study Design

This study was designed as a cross-sectional retrospective study and carried out at a single Private B&R Dental Center in Sulaymaniyah city, Kurdistan Region/Iraq in the period from 1 January 2020 to 30 June 2022. This study was approved by the Ethics Committee of the University of Sulaimani College Of Dentistry (No. 152/23; date 29 March 2023).

### 2.2. Sample Size Calculation

In this study, the sample size was calculated using the G*Power version 3.1.9.4 and a point biserial correlation model. The sample size for a correlation analysis is determined by the following formula:n = (Zα^2^ + Zβ^2^)/r^2^ + 3
where n is the total sample size; Zα is the Z-score for the significance level (α = 0.05, one-tailed: Zα = 1.645); Zβ is the Z-score for the desired power (1 − β = 0.95, Zβ = 1.645); r is the effect size or correlation coefficient (e.g., r = 0.1 for a small effect size); and +3 is the adjustment factor for the point biserial correlation.

The following parameters were used: test type—one-tailed point biserial correlation model; effect size (|r|)—0.1 (small effect size, as expected for this research); significance level (α)—0.05; power (1 − β)—0.95 (95% power); and clustering adjustment—each participant contributed 12 of their teeth. The design effect was calculated based on an intraclass correlation coefficient (ICC) of 0.1 to adjust for clustering.

In the design effect (DE), the total sample size was adjusted for clustering using the following formula:DE = 1 + (m − 1) × ICC
where m is the number of teeth per participant (12 participants), and ICC is the assumed intraclass correlation coefficient (0.1).

The effective sample size was then determined by dividing the calculated sample size by the design effect.

The resulting total sample size contained 1200 teeth, which equated to 100 participants contributing 12 teeth each.

### 2.3. Study Sample

A total of 1200 maxillary and mandibular teeth were selected from 100 CBCT images of 48 males and 52 females, with an average age of 34.69 years. This included 200 maxillary central incisors (11 and 21), 200 maxillary lateral incisors (12 and 22), 200 maxillary canines (13 and 23), 200 mandibular central incisors (31 and 41), 200 mandibular lateral incisors (32 and 42), and 200 mandibular canines (33 and 43). Patients underwent CBCT scans for various purposes, such as implant planning, evaluation of maxillofacial conditions, orthodontic purposes, oral surgery, and endodontic procedures. None of the CBCT scans were conducted solely for this study. All samples satisfied the following inclusion criteria: individual had to be over 18 years old and have all maxillary and mandibular anterior teeth present, which had to be in a sound state, with angle class I occlusion, with no rotation or malposition, with fully developed roots, and have CBCT images of good quality. The exclusion criteria included the following: images in which one or more anterior teeth were missing; implants in the anterior tooth area; distortions or poor quality; presence of pathological dento-alveolar conditions (e.g., cysts) that might cause abnormal bone remodeling; and teeth with root canal therapy, periapical lesions, and external root resorption.

### 2.4. CBCT Measurements

All CBCT images were captured using a Carestream 9600 (Carestream Dental, Marne-la-Vallée, France). The technical details were as follows: a 10 cm spherical imaging volume, an isotropic voxel size of 75 μm × 5 μm × 5 μm, and a field of view with a 10 cm diameter. The CBCT radiographs were taken with the following settings: 120 kV, 8 mA, and an exposure time of 20 s. Moreover, we utilized a CS 3D imaging software 1.9 (Carestream Dental) and the error margin of this software measuring tool was 3.71. The images were reviewed using the software’s built-in features in the axial plane, with adjustments made to contrast and brightness as necessary for optimal visualization.

Measurements on the CBCT scans were assessed by a radiologist. The arch form selector tool was positioned at the center of the arch in the axial plane. The curving of dental arches was performed manually in the Curve Slicing section using the Dental Arch Creation Mode in the Carestream 9600 CBCT software (CS Imaging Version 8). This involved tracing a panoramic curve along the midline of teeth to align with the natural contours of the upper and lower arches, which were analyzed separately.

After curving the arch, the alignment for each tooth was adjusted individually according to its long axis to ensure anatomical accuracy. This allowed for the generation of perpendicular slices that accurately represented the labial, palatal (or lingual), and alveolar bone measurements. Root position, angulation, and buccal and palatal/lingual bone thicknesses in both maxillary and mandibular anterior teeth were analyzed by examining cross-sectional images taken at the tooth’s midpoint, aligned parallel to its long axis. The positions of the maxillary and mandibular anterior tooth roots within the alveolar bone were determined based on the location of the apex as follows [14,15] (Figure 1 and Figure 2):

Class I: The root is located against the labial cortical plate.

Class II: The root is located centrally within the alveolar housing, not in contact with either the labial or the palatal cortical plate at the apical third.

Class III: The root is located against the palatal cortical plate.

Class IV: At least two-thirds of the root is in contact with both the labial and palatal cortical plates.

The buccal type was further categorized into subtypes I, II, and III. In subtype I, the incisor root was encased by the buccal bone wall, with the bone thickness increasing towards the apex. In subtype II, the incisor root was covered by a thinner buccal bone wall compared to subtype I, and the bone thickness did not significantly increase towards the apex, which was still covered by bone along the tooth’s long axis. In subtype III, the apex was angled strongly towards the buccal side, and it was not covered by bone along the tooth’s long axis [16] (Figure 3).

The thicknesses of the buccal and palatal/lingual bone walls of maxillary and mandibular anterior teeth were measured perpendicular to the long axis of the tooth at three specific points: 2 mm from crest, 4 mm from crest, and at the apical root apex [17] (Figure 4).

Additionally, the angle between the tooth’s long axis and the long axis of the related alveolar bone was measured. The long axis of the tooth was defined as the line extending from the lowest point of the crown to the highest point of the apex in the cross-sectional image [2] (Figure 5).

### 2.5. Reliability Test

For reliability all measurements were taken twice by the same observer with a two-week interval between recordings, and the average of these values was used for the analysis.

The intraclass correlation coefficient (ICC) is a value between 0 and 1, where values below 0.5 indicate poor reliability; values between 0.5 and 0.75 indicate moderate reliability; values between 0.75 and 0.9 indicate good reliability; and any values above 0.9 indicate excellent reliability [18].

### 2.6. Statistical Analysis

Kolmogorov–Smirnov test was used to assess the normality of the data. Descriptive statistics, including means, frequencies, and percentages, were calculated. As the data did not follow a normal distribution, the Kruskal–Wallis test was employed to compare the angulation between teeth and SRP classes. To identify differences between groups, Mann–Whitney U test was conducted. SPSS software (version 26.0, IBM, Armonk, NY, USA) was utilized to perform statistical analyses, with statistical significance set at *p* < 0.05.

## 3. Results

In this study, 100 patients (48 males and 52 females, with an average age of 34.69 years) were divided into the following three groups: group I (ages ≤ 25), group II (ages 26–40), and group III (ages ≥ 41).

The score from the analysis of intra-examiner reliability was 0.89, which indicates total dependability of the study.

In the current study, we examined SRP in relation to the surrounding bone structures. The average frequency distribution of SRP for maxillary anterior teeth showed that most were positioned buccally as Class I and subtype III, while mandibular anterior teeth were primarily classified as Class IV and subtype II, as shown in Table 1 and Table 2. 

The average sagittal angulation of maxillary anterior teeth ranged from 5.9 degrees for tooth 12 to 8.2 degrees for teeth 13 and 23. For mandibular anterior teeth, the angulation ranged from 7.4 degrees for tooth 33 to 10.3 degrees for tooth 41 (Table 3).

Statistical analysis was performed to analyze the effects of different parameters on the bone thickness and angulation using the Kruskal–Wallis test. Moreover, *p*-values equal to or less than 0.05 were regarded as significant statistical differences. For groups that showed significant difference based on the Kruskal–Wallis test, Mann–Whitney U test was performed to show the significance difference between the two groups.

The apical portion had higher means of bone thickness compared to the coronal and middle thirds in both buccal and palatal/lingual surfaces.

The Kruskal–Wallis test, performed on maxillary and mandibular arches, did not detect any significant differences in the bone thickness between right and left teeth (i.e., central incisor, lateral incisor, or canines) in the coronal, middle, and apical regions from buccal and palatal aspects (Table 4 and Table 5).

In the coronal third of maxillary teeth, there were no significant age-related differences in bone thickness of the buccal side ( *p*-value > 0.05, Kruskal–Wallis test), except in the coronal third of #21, which had the least bone thickness in group III (*p* < 0.05) (Figure 6A).

In the middle third, there was no significant difference related to age in the bone thickness of the buccal side of any maxillary anterior teeth (*p* > 0.05) (Figure 6A).

In the apical third, the bone thicknesses of the buccal sides of teeth #12, #22, and #23 indicated a significant difference related to age, with the highest bone thickness found in group 1 (*p* ≤ 0.05), whereas #13, #11, and #21 showed no significant difference related to age (*p*-value > 0.05) (Figure 6A).

Regarding the palatal surface, there was no age-related significant difference in the bone thickness of maxillary anterior teeth in the coronal third (*p*-value > 0.05), except for tooth #23, which had the highest bone thickness in group I (*p* < 0.05) (Figure 6B).

In the middle third, there was no age-related significant difference in the bone thickness of palatal sides of any maxillary anterior teeth (*p*-value > 0.05), except for #22, which showed a lower bone thickness in group 2 (*p* < 0.05) (Figure 6B).

In the apical third, there was no age-related significant difference in the bone thickness of maxillary anterior teeth (*p* > 0.05), except that #13 and #22 showed a lower bone thickness in group II (*p* < 0.05) (Figure 6B).

Regarding the angulation, there was an age-related statistical difference in the angulation of maxillary anterior teeth (*p* < 0.05), with group III showing the highest angulation, whereas #11 and #21 showed no statistical difference (*p* > 0.05) (Figure 6C).

In mandibular teeth, there was no age-related significant difference in the bone thickness of the buccal side in the coronal and middle thirds (*p*-value > 0.05, Kruskal–Wallis test) (Figure 7A).

In the apical third, there was a age-related significant difference in the bone thickness of the buccal side, with the highest bone thickness being found in group 1 (*p* < 0.05), whereas #42 and #43 showed no significant difference related to age (*p*-value > 0.05) (Figure 7A).

In the lingual surface, there was no age-related significant difference in bone thickness of mandibular anterior teeth in the coronal and middle third (*p*-value > 0.05) (Figure 7B).

In the apical third, the bone thickness of mandibular anterior teeth was significantly higher in group 3 (*p* < 0.05), except for #31, which exhibited no significant difference (*p* > 0.05) (Figure 7B).

Regarding the angulation, there was no age-related statistical difference in the angulation of mandibular anterior teeth (*p* > 0.05) (Figure 7C).

In maxillary teeth, there was no gender-related significant difference in the bone thickness of the buccal side in the coronal and apical thirds (*p*-value > 0.05, Kruskal–Wallis test) (Figure 8A).

In the middle third, the bone thicknesses in the buccal sides of all maxillary anterior teeth were significantly higher in males (*p* > 0.05) (Figure 8A).

Regarding the palatal surface, there was no gender-related significant difference in the bone thickness of maxillary anterior teeth in the coronal third (*p*-value > 0.05) (Figure 8B).

In the middle third, there was a gender-related significant difference in the bone thickness of the palatal side of all maxillary anterior teeth, with a higher bone thickness being noted in males (*p* < 0.05), except for #21 and #22, which showed no statistical difference (*p* > 0.05) (Figure 8B).

In the apical third, there was no gender-related significant difference in the bone thickness of maxillary anterior teeth in the apical third (*p* > 0.05), except for #13 and #12, which showed a lower bone thickness in the female group (*p* < 0.05) (Figure 8B).

Regarding the angulation, there was a gender-related statistical difference in angulation of maxillary anterior teeth (*p* < 0.05), with the male group showing the highest angulation in #13, #12, and #23, while teeth #11, #21, and #22 showed no statistical difference (*p* > 0.05) (Figure 8C).

In mandibular anterior teeth, there was no gender-related significant difference in the bone thickness of the buccal side in the coronal and apical thirds (*p*-value > 0.05, Kruskal–Wallis test). Although higher bone thickness was found in the apical third of females compared to the male group, the difference was not statistically significant (*p*-value > 0.05) (Figure 9A).

In the middle third, the bone thickness in the buccal side of mandibular anterior teeth was significantly higher in males in #33, #42, and #43 (*p* > 0.05), while #32, #31, and #41 showed no significant difference (Figure 9A).

In the lingual surface, there was no gender-related significant difference in bone thickness of mandibular anterior teeth in the coronal and middle thirds (*p*-value > 0.05) (Figure 9B).

In the apical third, males had more bone thickness in the palatal side compared to females, with significant differences being noted in #33, #32, #42, and #43 (*p* < 0.05), while in #31 and #41 this difference was not statistically significant (*p* > 0.05) (Figure 9B).

Regarding the angulation, there was no gender-related significant difference in angulation of mandibular anterior teeth (*p* > 0.05) (Figure 9C).

## 4. Discussion

In this study, we investigated SRP in maxillary and mandibular anterior teeth, instead of other areas of alveolar bone, for the following reasons: immediate implant placement in the maxillary and mandibular anterior areas presents unique challenges, including a higher risk of complications and potential bone perforation during the procedure [11,19,20]. Furthermore, labial bone perforation is most commonly observed in the mandibular central incisors, which can jeopardize the success of the implant in the long term [21]. Therefore, understanding the anatomical characteristics of this area is essential for effective treatment planning and improved patient outcomes.

Additionally, understanding the ridge profile is crucial for achieving precise three-dimensional implant placement. Currently, CBCT technology and its associated software facilitate a straightforward evaluation of the ridge profile, including measurements of the overall, facial, and palatal thicknesses, as well as the concavity of the facial bone plate [17].

To the author’s knowledge this is the first study to measure the thickness and angulation of maxillary and mandibular teeth in the buccal and palatal/lingual regions in relation to age and gender using CBCT.

Our findings indicated that the frequency of SRP in the maxillary anterior region across all teeth could be classified as Class I and buccal subtype Class III. Notably, Class III had the lowest prevalence within the Iraqi subpopulation and this result is consistent with the results of previous studies [22,23,24]. Therefore, SRP Class I is considered a favorable condition for immediate implant placement. This implies that if guidelines are adhered to, most cases in the maxillary anterior region could qualify for immediate implant placement.

In contrast, mandibular anterior teeth typically have thin facial and lingual crests, which makes the thickness of the apical lingual bone crucial for immediate implant placement. While support can be derived from the lingual bone, the angulation and types of teeth influence the planning for immediate implant placement [25]. In this study, Class IV, in which at least two-thirds of the root are engaging both the labial and palatal cortical plates, was the most prevalent category among all mandibular anterior teeth. Furthermore, this offers a better prognosis for immediate implant placement, irrespective of bone thickness, as both the buccal and lingual bone walls are present [14,26]. These findings are comparable with other studies conducted by Moghaddas and Behravan [27], which reported that 16.3% of the cases were classified as Class IV, whereas Ikbal et al. [21] reported that Class IV SRP was the most prevalent in the right and left lateral incisors by 11.2% (these differences can be attributed to ethnic differences among these studies).

In this study, subtypes III were more frequent in the maxillary anterior teeth. Other researchers have observed similar findings [22,28]. The buccal subtype III teeth featured a very thin buccal plate, and the root apex was angled significantly toward the buccal side, extending in front of the natural contour of the maxillary alveolar bone. As a result, patients with a subtype III root position might not be ideal candidates for immediate implant placement [2,26].

The average sagittal angulation of the maxillary anterior teeth ranged from approximately 5.9 degrees for maxillary right lateral incisors to 8.2 degrees for maxillary right and left canines. Moreover, for mandibular anterior teeth, the angulation varied from 7.4 degrees for mandibular right canine to 10.3 degrees for mandibular right central incisors. Similar results were obtained from other studies [4,9,29]. The angle was higher around canines than other incisor teeth. Understanding the angles of teeth relative to the alveolar bone will help practitioners prevent complications like perforation and fenestration during implant placement, while also ensuring long-lasting esthetic success.

In the anterior maxilla, the mean labial plate thickness ranged from 0.63 to 1.71 mm, and the palatal plate thickness ranged from 0.66 to 8.20 mm. Moreover, in the anterior mandible, the thickness of labial and lingual plates ranged from 0.50 to 3.90 mm and from 0.47 to 3.51 mm, respectively. These findings agree with other studies by Shafizadeh et al. [30] and Tsigarida et al. [31]

This study also investigated the relationship between age and thickness/angulation of the alveolar bone in the maxillary and mandibular anterior teeth. We found significant differences among age groups in bone thickness of all maxillary and mandibular tooth apexes; bone thickness decreases with advancing age. The study results of Gakonyo et al. [32], Somvasoontra et al. [33], Almahdi and Alasqah [34], and Rodrigues et al. [23] were similar to our study results regarding age. However, Nowzari et al. [35] and Aljabr et al. [36] found no difference in their age group comparison (below 30 years and above 30 years).

Gender variation in the buccal and palatal/lingual bone thickness was evaluated in the present study. We found significant differences in the middle thirds of both buccal and palatal bone thicknesses; reduced bone thickness in females for all anterior teeth; and reduced bone thickness in the middle third of labial bone thickness of most mandibular anterior teeth. These findings were corroborated by Januário et al. [7], Nowzari et al. [35], Wang et al. [3], Ozdemir et al. [37], and Srebrzyńska-Witek et al. [4], all of whom observed that females had lower bone thickness than males. On the other hand, other studies [5,17,23] registered no gender differences between SRP and alveolar bone thickness. Ohiomoba et al. [38] found that body mass index and age are positively linked to bone thickness and density. Similar to findings related to gender, some studies have reported variations in bone thickness between younger and older age groups, while others found no differences. Variations in sample sizes and parameters selected in CBCT studies may explain the discrepancies observed in these findings.

There are some limitations to this study. Firstly, discrepancies between the present study and others may have been due to the small sample size and some of the findings might not be generalizable to other ethnic groups. Secondly, different dento-skeletal relationships between ethnic groups may play a role in ridge profiles of the anterior maxilla and mandible. In addition, bone thickness is influenced by the orientation of teeth within the arch, as well as tooth shape, root structure, and other morphological characteristics; future studies could include these factors to provide clearer insights and improve correlations among analyzed teeth. Furthermore, the error margins of this study is 3.71% with a mean percentage error of 9.44% and standard deviation of 7.21%. This limitation points to uncertainty in software accuracy and implies caution when translating measurements into clinical decisions or research findings.

## 5. Conclusions

It can be concluded that the SRP type 1 is most typically present in the anterior maxillary teeth and type IV in anterior mandibular teeth. The maxillary anterior teeth were placed at an angle ranging from 5.9–8.2 degrees, while the mandibular incisors were placed parallel to the alveolar ridge. Bone perforations may be reduced by using taper implants and abutments at an angle of 5°–10° for maxillary anterior teeth, while straight implants may be preferred for mandibular anterior teeth. Bone thickness significantly decreased with age in all maxillary and mandibular anterior teeth at the apex, while males had significantly more bone thickness compared to females in the middle thirds of all maxillary and mandibular anterior teeth.

## Figures and Tables

**Figure 1 diagnostics-14-02756-f001:**
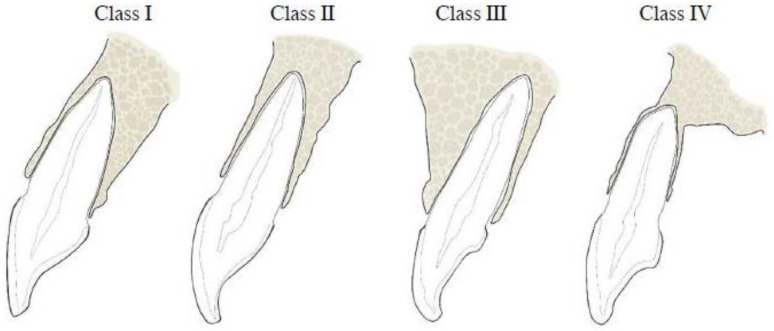
Schematic diagram of the SRP classification of maxillary anterior teeth as reported by Kan et al. [14].

**Figure 2 diagnostics-14-02756-f002:**
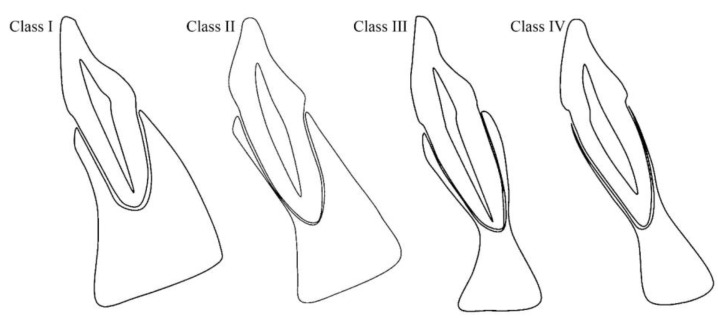
Schematic diagram of the SRP classification of mandibular anterior teeth as reported by Zhang et al. [15].

**Figure 3 diagnostics-14-02756-f003:**
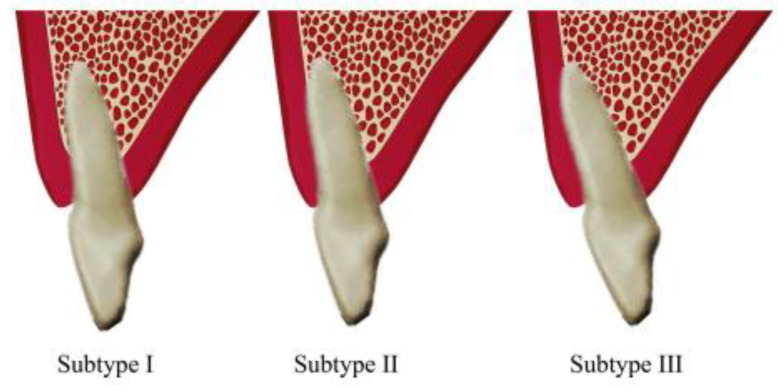
Schematic diagram of the buccal-type sub-classification by Xu et al. [16].

**Figure 4 diagnostics-14-02756-f004:**
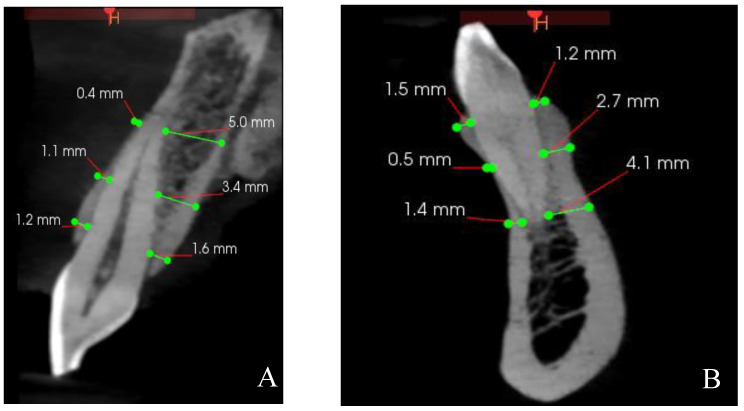
The thicknesses of the buccal and palatal/lingual bones were measured at 2 mm from crest, 4 mm from crest, and at the root apex. (**A**) Maxillary tooth. (**B**) Mandibular tooth.

**Figure 5 diagnostics-14-02756-f005:**
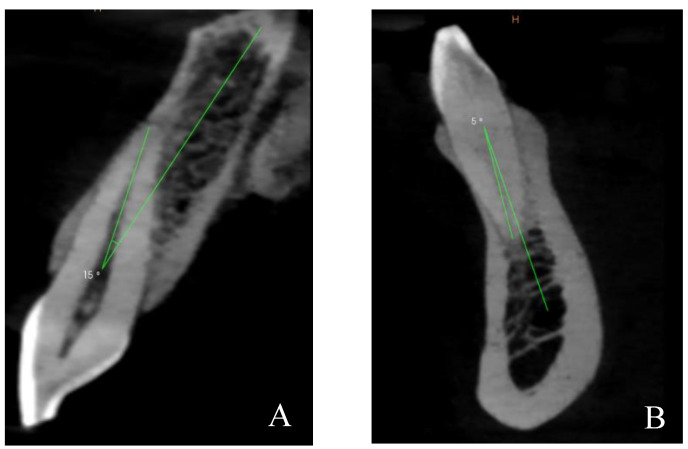
The angle between the long axis of the tooth and the long axis of the corresponding alveolar bone was measured. (**A**) Maxillary tooth. (**B**): Mandibular tooth.

**Figure 6 diagnostics-14-02756-f006:**
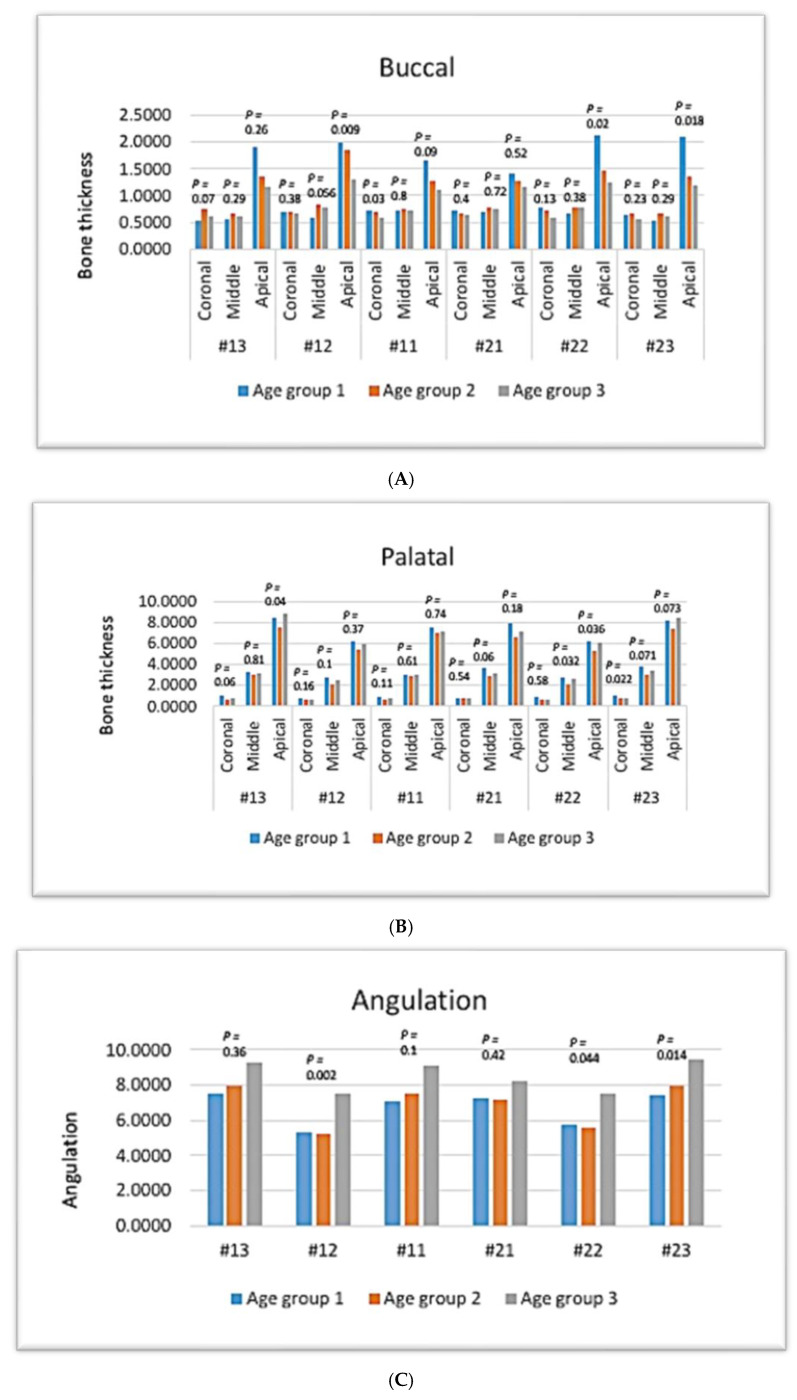
The effect of age on maxillary anterior teeth. Bone thickness: (**A**) buccal, (**B**) palatal, and (**C**) angulation.

**Figure 7 diagnostics-14-02756-f007:**
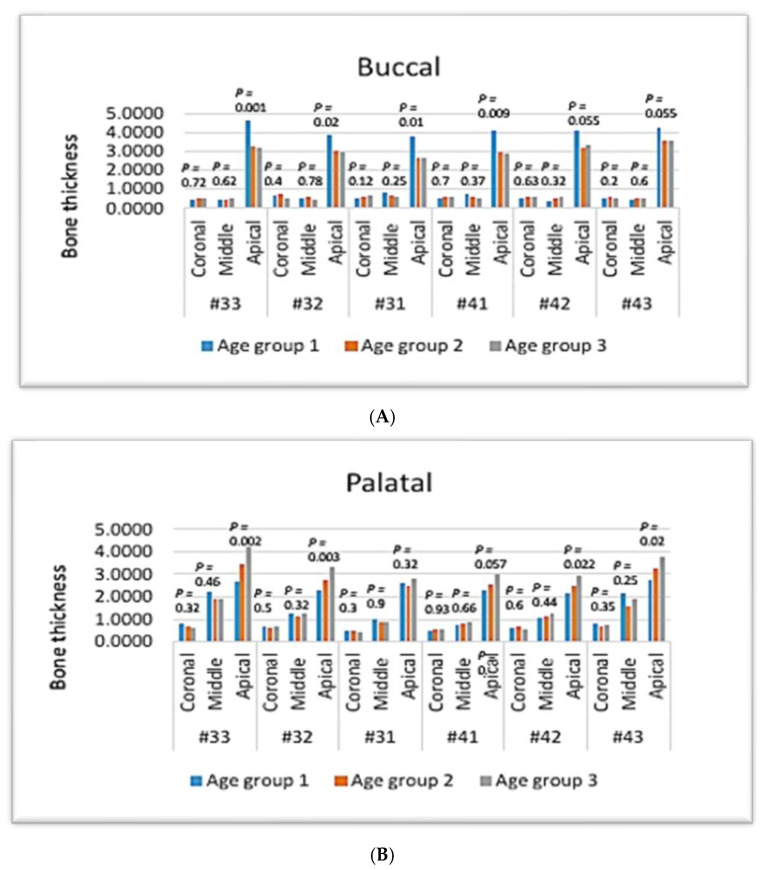
The effect of age on mandibular anterior teeth. Bone thickness: (**A**) buccal, (**B**) palatal, and (**C**) angulation.

**Figure 8 diagnostics-14-02756-f008:**
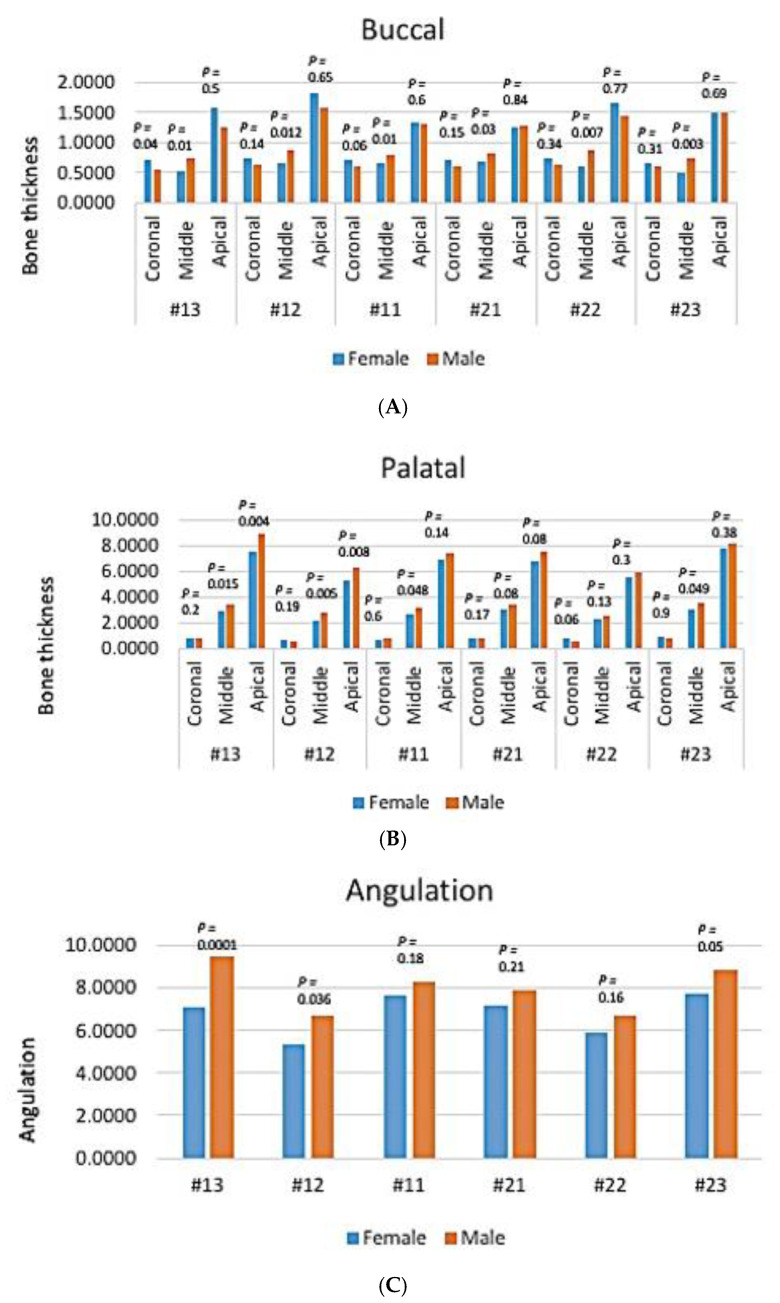
The effect of gender on maxillary anterior teeth. Bone thickness: (**A**) buccal, (**B**) palatal, and (**C**) angulation.

**Figure 9 diagnostics-14-02756-f009:**
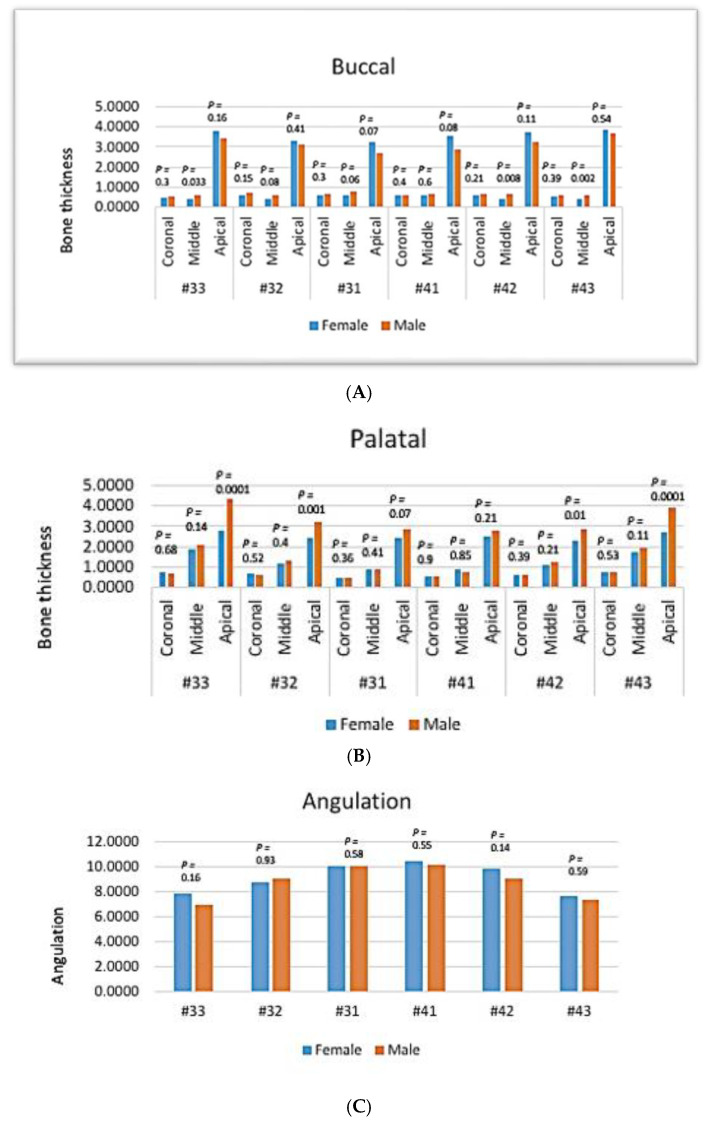
The effect of gender on mandibular anterior teeth. Bone thickness: (**A**) buccal, (**B**) palatal, and (**C**) angulation.

**Table 1 diagnostics-14-02756-t001:** Frequencies of classifications and sub-classifications of SRP in maxillary anterior teeth.

Sagittal Root Position	Frequencies in Percentages						
	Tooth	13#	12#	11#	21#	22#	23#
Classification	Class I	63%	69%	90%	85%	76%	84%
	Class II	35%	29%	10%	15%	22%	14%
	Class III	0%	1%	0%	0%	0%	0%
	Class IV	2%	1%	0%	0%	2%	2%
	Total	100%	100%	100%	100%	100%	100%
Sub-classification	Subtype I	34%	14%	13%	8%	8%	3%
	Subtype II	5%	12%	7%	7%	13%	10%
	Subtype III	61%	74%	80%	85%	79%	87%
	Total	100%	100%	100%	100%	100%	100%

**Table 2 diagnostics-14-02756-t002:** Frequencies of classifications and sub-classifications of SRP in mandibular anterior teeth.

Sagittal Root Position	Frequencies in Percentages						
	Tooth	33#	32#	31#	41#	42#	43#
Classification	Class I	7%	5%	2%	3%	1%	10%
	Class II	10%	7%	3%	8%	5%	13%
	Class III	7%	4%	1%	2%	2%	3%
	Class IV	76%	84%	94%	87%	92%	74%
	Total	100%	100%	100%	100%	100%	100%
Sub-classification	Class I	26%	14%	31%	37%	37%	26%
	Class II	54%	74%	62%	47%	56%	57%
	Class III	20%	12%	7%	16%	7%	17%
	Total	100%	100%	100%	100%	100%	100%

**Table 3 diagnostics-14-02756-t003:** Descriptive statistics of angulations of maxillary and mandibular teeth.

	Tooth Type											
	#13	#12	#11	#21	#22	#23	#33	#32	#31	#41	#42	#43
Mean	8.2	5.9	7.9	7.5	6.2	8.2	7.4	8.9	10.0	10.3	9.4	7.5
Median	8.0	6.0	8.0	8.0	6.0	8.0	8.0	9.0	10.0	10.0	10.0	8.0
St. Dev.	2.86	3.1	3.7	3.6	3.3	2.9	2.8	3.3	4.03	3.6	2.9	2.6
Range	15.0	18.0	23.0	21.0	20.0	17.0	17.0	17.1	19.5	17.0	14.0	12.0

**Table 4 diagnostics-14-02756-t004:** Descriptive statistics of bone thicknesses on the buccal surfaces of both the maxillary and mandibular teeth.

		Tooth Type											
		#13	#12	#11	#21	#22	#23	#33	#32	#31	#41	#42	#43
Coronal	Mean	0.64	0.69	0.66	0.66	0.69	0.63	0.50	0.63	0.61	0.58	0.59	0.55
	Median	0.60	0.60	0.70	0.70	0.65	0.60	0.50	0.60	0.60	0.60	0.60	0.60
	St. Dev.	0.45	0.35	0.29	0.29	0.33	0.31	0.31	0.40	0.34	0.37	0.33	0.30
	Range	2.90	1.70	1.60	1.70	1.60	1.50	1.60	2.30	1.50	1.60	1.30	1.20
Middle	Mean	0.63	0.75	0.73	0.74	0.74	0.61	0.47	0.50	0.67	0.60	0.52	0.49
	Median	0.50	0.60	0.70	0.70	0.60	0.55	0.30	0.40	0.50	0.50	0.40	0.40
	St. Dev.	0.46	0.62	0.33	0.35	0.57	0.39	0.45	0.43	0.52	0.45	0.41	0.37
	Range	2.10	3.60	1.60	2.00	2.80	1.90	2.60	2.30	3.0	2.10	1.90	2.00
Apical	Mean	1.43	1.71	1.31	1.27	1.55	1.49	3.61	3.22	2.98	3.22	3.48	3.76
	Median	1.25	1.50	1.15	1.05	1.30	1.30	3.75	3.20	3.05	3.00	3.40	3.90
	St. Dev.	1.15	1.22	0.86	0.80	1.10	1.17	1.54	1.34	1.54	1.57	1.38	1.33
	Range	7.20	8.40	5.40	4.40	8.40	6.80	7.30	7.80	9.90	9.40	7.70	8.20

**Table 5 diagnostics-14-02756-t005:** Descriptive statistics of bone thicknesses on palatal/lingual surfaces of both the maxillary and mandibular teeth.

		Tooth Type											
		#13	#12	#11	#21	#22	#23	#33	#32	#31	#41	#42	#43
Coronal	Mean	0.78	0.66	0.74	0.79	0.68	0.85	0.70	0.65	0.47	0.51	0.63	0.74
	Median	0.70	0.60	0.70	0.75	0.60	0.80	0.70	0.60	0.40	0.45	0.60	0.70
	St. Dev.	0.51	0.40	0.36	0.37	0.53	0.66	0.33	0.37	0.27	0.32	0.37	0.33
	Range	3.30	3.20	1.90	2.10	4.80	5.90	2.40	2.90	1.10	1.90	3.10	1.90
Middle	Mean	3.14	2.42	2.95	3.18	2.44	3.33	1.99	1.21	0.91	0.83	1.14	1.85
	Median	3.20	2.30	3.00	3.10	2.35	3.20	1.90	1.10	0.70	0.60	1.10	1.70
	St. Dev.	1.12	1.09	1.18	1.18	1.10	1.29	0.98	0.72	0.68	0.58	0.64	1.09
	Range	6.80	5.00	6.20	5.50	6.50	7.0	4.50	3.0	4.60	2.60	3.10	7.30
Apical	Mean	8.20	5.84	7.16	7.15	5.77	7.95	3.51	2.79	2.64	2.62	2.54	3.30
	Median	8.25	5.75	7.0	6.85	5.50	8.0	3.30	2.75	2.70	2.50	2.45	3.30
	St. Dev.	2.48	1.96	2.16	2.24	1.86	2.23	1.84	1.19	1.26	1.12	1.05	1.41
	Range	12.7	9.80	10.2	12.5	8.0	10.0	9.40	6.10	8.30	5.30	5.70	6.80

## Data Availability

Request to the corresponding author.

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
