# Peer review of "Analysis of the Sagittal Root Position of the Maxillary and Mandibular Anterior Teeth in the Alveolar Bone Using Cone-Beam Computed Tomography"

_diagnostics, 2024, doi:10.3390/diagnostics14232756_

Round 1

Reviewer 1 Report

Comments and Suggestions for Authors

The authors have analyzed the sagittal root position of the maxillary and mandibular anterior teeth in the alveolar bone using cone-beam computed tomography.

The study methodology can include how the arch was drawn and sectioning was performed. The pictorial representation of the classification does not match with the description provided for mandibular anteriors. What was the ICC score for inter and intra-observer analysis?

Discussion section can include systematic reviews performed on the similar topic.

Author Response

Reviewer 1

The author would like to thank the reviewer for this thoughtful and meaningful suggestions that surely helped me to further improve this manuscript. Based on the reviewer’s suggestions, I have revised the manuscript and fully responded to all the Referee’s comments. I have highlighted the changes to the manuscript and easy to track. Please find a point-by-point response to the reviewer’s comments below:

  1. The study methodology can include how the arch was drawn and sectioning was performed.

Author’s Response: Many thanks for this thoughtful comment. This has been added accordingly in (methods section; 2.3 CBCT measurement section (line 135-144)). Thanks again.

  1. The pictorial representation of the classification does not match with the description provided for mandibular anteriors

Author’s Response: This has been revised and highlighted in discussion section (line 535-539). Many thanks.

  1. What was the ICC score for inter and intra-observer analysis?

Author’s Response:

This has been amended and highlighted accordingly in methodology by adding section (2.4 Reliability test (line 187-192) and score was mentioned in result section (226-227).

  1. Discussion section can include systematic reviews performed on the similar topic.

Author’s Response:

Thanks to this thoughtful suggestion. This has been added to discussion section (line 558-562 ). Thanks again.

  • doi: 10.1016/j.ortho.2021.07.002. Epub 2021 Aug 6. PMID: 34366263.
  • doi: 10.1111/jcpe.13347. Epub 2020 Sep 16. PMID: 32691437.

Finally, the author would like to thank the reviewer again for the constructive feedback on this article that surely helped to improve the contents and quality. I believe that the quality of the revised manuscript has been improved remarkably and it will be acceptable for publication.

Yours Sincerely,

Reviewer 2 Report

Comments and Suggestions for Authors

The authors are commended for the efforts in developing the current manuscript.

However the following questions are important to clarify to improve the quality of the manuscript.

1. How was the sample size calculated?

2. Were the cbct obtained solely for this study?

3. Was the examiner calibrated? What was the intra-reliability test value?

4. What was the margin of error using the software measuring tool? Is there a limitation using this measuring tool associated with the software? This should be included in the limitations.

5. Figure 6 is not necessary since the age description was described in test already

Author Response

Reviewer 2

The author would like to thank the reviewer for this thoughtful and meaningful suggestions that surely helped me to further improve this manuscript. Based on the reviewer’s suggestions, I have revised the manuscript and fully responded to all the Referee’s comments. I have highlighted the changes to the manuscript and easy to track. Please find a point-by-point response to the reviewer’s comments below:

  1. How was the sample size calculated?

Author’s Response: This has been added and highlighted to methodology under (2.2 Sample size calculation section) (line 83- 107). Thanks for this thoughtful comment.

  1. Were the cbct obtained solely for this study?

Author’s Response: Thank you for these comment. Patients has these CBCTs was taken for a variety of reasons and not solely for the purpose of this study. This has been added and highlighted in methodology in (2.3. Study Sample (line 113-116).

  1. Was the examiner calibrated? What was the intra-reliability test value?

Author’s Response: This has been amended and highlighted accordingly in methodology by adding section (2.4 Reliability test (line 187-192) and score was mentioned in result section (226-227). Thanks for this comment.

  1. What was the margin of error using the software measuring tool? Is there a limitation using this measuring tool associated with the software? This should be included in the limitations.

Author’s Response: I am grateful for this thoughtful comment. The margin of error using the software measuring tool was (3.71) and this was added to section (2.4. CBCT measurement line (130-131) and also added as limitation in discussion section (line 591-594).

The margin of error was calculated according to:

Mean of percentage of error

9.44

SD

7.21

Critical t value (0.05-16)

2.12

Margin of error

3.71

0.05 is the significance level for a 95% confidence interval (1−0.95=0.05).

16 is the degrees of freedom (n−1n - 1n−1).

Margin of Error=t SE

  1. Figure 6 is not necessary since the age description was described in test already.

Author’s Response: This has been amended accordingly.

Finally, the author would like to thank the reviewer again for the constructive feedback on this article that surely helped to improve the contents and quality. I believe that the quality of the revised manuscript has been improved remarkably and it will be acceptable for publication.

Yours Sincerely,

Round 2

Reviewer 2 Report

Comments and Suggestions for Authors

Dear Authors, 

Thank you for addressing the comments and adjusting the manuscript.